# Advances in Enzymatic Synthesis of D-Amino Acids

**DOI:** 10.3390/ijms21093206

**Published:** 2020-05-01

**Authors:** Loredano Pollegioni, Elena Rosini, Gianluca Molla

**Affiliations:** Department of Biotechnology and Life Sciences, University of Insubria, via J.H. Dunant 3, 21100 Varese, Italy; loredano.pollegioni@uninsubria.it (L.P.); elena.rosini@uninsubria.it (E.R.)

**Keywords:** D-amino acids, biocatalysis, cascade reactions, protein engineering, stereoselective reactions

## Abstract

In nature, the D-enantiomers of amino acids (D-AAs) are not used for protein synthesis and during evolution acquired specific and relevant physiological functions in different organisms. This is the reason for the surge in interest and investigations on these “unnatural” molecules observed in recent years. D-AAs are increasingly used as building blocks to produce pharmaceuticals and fine chemicals. In past years, a number of methods have been devised to produce D-AAs based on enantioselective enzymes. With the aim to increase the D-AA derivatives generated, to improve the intrinsic atomic economy and cost-effectiveness, and to generate processes at low environmental impact, recent studies focused on identification, engineering and application of enzymes in novel biocatalytic processes. The aim of this review is to report the advances in synthesis of D-AAs gathered in the past few years based on five main classes of enzymes. These enzymes have been combined and thus applied to multi-enzymatic processes representing in vitro pathways of alternative/exchangeable enzymes that allow the generation of an artificial metabolism for D-AAs synthetic purposes.

## 1. Introduction

D-amino acids (D-AAs) are still considered the “unnatural” enantiomers of amino acids because they are not used in protein synthesis [1]. The interest in D-AAs is related to their occurrence in nature as free molecules or as components of biomolecules, their biological properties and peculiar biosynthetic pathways, and for the expansion of analytical techniques for their quantification [2]. Although D-AAs do not exist in the nature as widely as their L-amino acids (L-AAs), naturally occurring D-AAs possess different and specific functions in different organisms (from a structural role as components of the peptidoglycan in the bacterial cell wall to act as neuromodulators in the mammalian brain) [3,4,5], are present in a variety of foodstuffs (affecting taste and flavor), and are responsible of antimicrobial and antiaging effects [6].

D-AAs are increasingly becoming important building blocks to produce pharmaceuticals and fine chemicals. D-AAs are currently used as key components in β-lactam antibiotics, fertility drugs, anticoagulants and pesticides; the semisynthetic antibiotics ampicillin and amoxicillin, which contain D-phenylglycine and D-*p*-Hydroxy-phenylglycine, respectively, are produced on a scale >5000 tons per year worldwide [7]. Furthermore, more than 20 D-AAs are produced at pilot- or full-scale levels.

During the years, a number of enzymes producing or metabolizing D-AAs have been discovered and devised to produce D-AAs [8,9]. While each of these methods are useful, they often suffer from various drawbacks including modest yields and enantioselectivity, low reaction rates, low titer of starting material, and the demand for multiple steps. These weaknesses represented the driving force that pushed numerous research groups worldwide to identify, engineer and apply enzymes in novel biocatalytic processes aimed to generate D-AAs by more-sustainable processes. 

The aim of this review is to report the advances in enzymatic synthesis of D-AAs reported in the past few years. The presentation is based on the five main classes of enzymes used, even if they have been frequently applied jointly in cascade, multi-enzymatic processes. 

## 2. Aminotransferases

Transaminases, also named aminotransferases (EC 2.6.1.X), catalyze the transfer of an amino group between an amino acid (donor) and a α-keto acid (acceptor) using a pyridoxal 5′-phosphate cofactor (PLP) [10].

### 2.1. D-Amino Acids Synthesis

In a pioneering work, the group of Kenji Soda reported the use of D-amino acid aminotransferase (DAAT, EC 2.6.2.21) to generate D-AAs from the corresponding α-keto acids and ammonia by coupling four enzymes [11]. In addition to DAAT from *Bacillus subtilis* sp. YM-1, which employs D-glutamate as amino donor, glutamate racemase from *Pediococcus pentosaceus* (to convert L-glutamate into the D-enantiomer), commercial glutamate dehydrogenase (to generate L-glutamate from α-ketoglutarate and ammonia) and commercial formate dehydrogenase (to regenerate NADH) were used (Figure 1A). D-valine, D-alanine, D-leucine, D-methionine D-aspartate and D-aminobutyrate have been synthesized from the corresponding α-keto acid with a >80% yield.

An alternative approach was recently used to generate different tryptophan derivatives by Parmeggiani [12] (Figure 1B). D-Tryptophan derivatives are important precursors of pharmaceuticals and natural products, such as tadalafil, lanreotide acetate, skyllamycin, metalloprotease inhibitors for pain treatment, prenylated tryptophans, inhibitors of breast cancer resistance protein, etc. In this process, a three-enzymatic system was set up coupling the synthesis of L-tryptophan derivatives from indoles by a tryptophan synthase from *Salmonella enterica* with the stereoinversion of the L-enantiomer into the D-AA by the oxidative deamination due to L-amino acid deaminase (LAAD, EC 1.4.3.2) from *Proteus myxofaciens* (PmaLAAD) and its transamination by a stereoselective D-aminotransferase variant from *Bacillus* sp. YM-1 (the T242G variant engineered to be active on various D-tryptophan derivatives). A total of 12 products containing electron-donating or withdrawing substituents at all benzene-ring positions on the indole group were produced, with a conversion yield in the 81–99% range, an isolation yield in the 63–70% range and an ee frequently >99%. This process was used at a preparative scale (5 mmol of D-tryptophan corresponding to 1.02 g).

By using a bi-enzymatic system, the cheap and available natural amino acid L-methionine was converted into D-homoalanine (Figure 1C) [13]. At first, L-methionine γ-lyase from *Fusobacterium nucleatum* catalyzed the conversion of L-methionine to 2-oxobutyrate, which was then aminated using D-alanine as amino donor by the DAAT from *Bacillus* sp. into D-homoalanine with a 90% ee and 87.5% conversion yield. The authors opted for the use of lyophilized whole cell systems.

While α-transaminases act on the α-amino groups, ω-transaminases abstract an amino group from a non-α position or even from primary amines that do not contain a carboxy group. ω-Transaminase from *Vibrio fluvialis JS17* was used to convert 3-fluoropyruvate into D-3-fluoroalanine using (*S*)-α-methylbenzylamine (α-MBA) as the amino donor; the process was hampered by a strong inhibition by acetophenone (the product of α-MBA deamination) [14]. This drawback was solved using a biphasic reaction (water/isooctane): under optimized conditions, D-3-fluoroalanine was produced with 70% conversion and ee >99% using 40 mM racemic α-MBA, 30 mM 3-fluoropyruvate and 3 U/mL enzyme. A different approach was used coupling the reaction of DAAT from *Bacillus sphericus* starting from the α-keto acid and D-alanine to generate the corresponding D-AA and iminopyruvate, with a variant of ω-transaminase from *Arthrobacter* sp. (ARTA) that converted the latter into D-alanine [15]. Using 450 mM iminopyruvate and 20 mM D-alanine, 2.02 g of D-phenylglycine were produced with 89% yield and ee >99%. Subsequently, the same group investigated the use of two (*R*)-selective ω-transaminases from *Aspergillus terreus* and *Aspergillus fumigatus* in the asymmetric synthesis of D-AAs from α-keto acids [16]. Such enzymes showed the highest amino donor reactivity for α-MBA, the absence of inhibition by acetophenone and the efficient use of α-keto acids corresponding to D-alanine, D-homoalanine, D-fluoroalanine, D-serine and D-norvaline. The latter D-AAs were produced with ee >99% and conversion yields in the 40–99% range (employing 60 mM racemic α-MBA, 20 mM α-keto acid, 3 U/mL ω-transaminase and 0.1 mM PLP).

### 2.2. Resolution of Racemic Mixtures

Since D-enantiomers are frequently more expensive than the corresponding L-AAs, stereoinversion represented a suitable way to generate D-AAs. On this side, D-phenylalanine was generated using an *Escherichia coli* W14(pR15ABK) strain selected for its ability to produce L-phenylalanine and for overexpressing the DAAT from *B. subtilis* W600 [17]: D-phenylalanine production reached 1.73 g/L in a 15 L fermenter.

Most recently, stereoinversion and deracemization of phenylalanine derivatives containing electron-donating or withdrawing substituents at different positions on the phenyl ring were performed using LAAD from *Proteus mirabilis* (PmLAAD), to convert the L-AA into the corresponding α-keto acid, followed by an engineered D-selective aminotransferase from *Bacillus* sp. YM-1 (the variant harboring the T242G substitution showed the best performance), (Figure 2A) [18]. The conversion was carried out using two *E. coli* cell strains overexpressing separately the two enzymes, employed as a whole-cell system, and 12 different L-phenylalanines. D-phenylalanine derivatives were synthesized with high enantiomeric excess (from 90% to >99%) from commercially available racemic mixtures or L-AAs. The process was also used to a preparative-scale: 76.9 mg of D-4-fluorophenylalanine was produced with ee >99% and an isolated yield of 84%. D-phenylalanine derivatives are used as chiral building blocks in the synthesis of important drugs, i.e., antibiotics, antidiabetics, and chemotherapeutic agents. The biocatalytic approach represents a valid alternative to asymmetric Strecker reactions because of the high enantioselectivity, as well as to fermentation processes that allow producing few substituted D-phenylalanine derivatives.

In recent years, a number of authors used an amino acid dehydrogenase coupled to a D-selective ω-transaminase to perform the deracemization and stereoinversion of racemic mixtures of amino acids (Figure 2B). Recombinant alanine dehydrogenase from *B. subtilis* and recombinant NADH oxidase from *Lactobacillus brevis* (required to recycle NADH and produced in *E. coli*) were coupled to the engineered ARTA in a one-pot reaction on 10 mM racemic alanine, serine or leucine, reaching a 80–99% ee [19]. Under optimized conditions, 100 mM D,L-alanine was deracemized in 24 h, using 1 mM NAD^+^ and 100 mM isopropylamine (>99% ee and 52% isolation yield).

## 3. D-Amino Acid Dehydrogenases

### 3.1. Synthesis of D-Amino Acids

The pioneering protein engineering work of Scott Novick [20] generated a highly stereoselective D-amino acid dehydrogenase (DAADH, EC 1.4.99.1) starting from a *meso*-diaminopimelate D-dehydrogenase (*m*-DAPDH) from *Corynebacterium glutaricum*. Three rounds of mutagenesis allowed the generation of the BC621 variant harboring the D155G/Q151L/R196M/T170I/H245N substitutions, able to produce D-AAs by reductive amination of the corresponding α-keto acids using NADPH and ammonia. The enzyme showed a broad substrate preference and was used to generate various D-AAs with >95% ee. Later on, the same enzyme was used to convert 4-Br-phenylpyruvic acid into D-4-Br-phenylalanine (40–70% isolation yield and ee ranging 74–99%), which was subsequently coupled with a panel of arylboronic acids to give D-biarylalanine derivates (Figure 3A), useful in synthesis of inhibitors of botulinum toxin, of amyloid-β-peptide aggregation and of kinesin-14 motor protein KIFC1 [21].

Alternative DAADHs were generated in recent years. The *m*-DAPDH from the thermophilic *Symbiobacterium thermophilum* bacterium showed a reductive amination activity on various α-keto acids [22]. Site-saturation mutagenesis at positions P146, T171, R181, H227 identified the H227V variant showing a specific activity of 2.4 µmol/min/mg protein on phenylpyruvic acid, corresponding to a >35-fold increase compared to the wild-type enzyme [23]. An increased activity on bulky substrates was observed for the W121L/H227I DAADH variant, e.g., 24- and 70-fold toward 2-oxo-4-phenylbutyric acid and phenylpyruvic acid, respectively. D-phenylalanine was generated with >99% ee and 85% yield [24]. A multi-enzymatic system made of L-threonine ammonia lyase from *E. coli*, DAADH and formate dehydrogenase (from *Pseudomonas* sp. *101* or *Candida boidinii*) was set up to produce D-2-aminobutyric acid using L-threonine (Figure 3B). D-2-aminobutyric acid is used in the synthesis of antibiotics, antiproliferatives and inhibitors of angiotensin-converting enzyme 2, brain-permeable polo-like kinase-2 and matrix metalloproteinase. At 50 mL reaction scale, 200 mM L-threonine and 300 mM sodium formate generated D-2-aminobutyric acid in 20 h, with >90% yield and >99% ee, with no need for external ammonia supplement [25].

An alternative DAADH is the *m*-DAPDH from *Ureibacillus thermosphaericus* strain A1 variant containing five substitutions (introduced by site-saturation mutagenesis) that was used to synthesize branched-chain D-AAs coupling DAADH and glucose oxidase/glucose system for NADPH recycling [26]. After 2 h at 65 °C and pH 10.5, 2-oxo-4-methylvaleric acid was converted to D-leucine with a 99% yield and >99% ee. This system was used to produce labeled D-AAs, namely D-[1-^13^*C*,^15^*N*]leucine, D-[1-^13^*C*]leucine, D-[^15^*N*]leucine, D-[^15^*N*]isoleucine, and D-[^15^*N*]valine. The D94A substitution prevented the steric clash with the side chain of large substrates; specific activity on phenylpyruvate (16.1 µmol/min/mg at 50 °C) was 8-fold higher than that of the wild-type DAADH [27]. For a review concerning the protein engineering and the structural studies on thermostable artificial DAADH, see [28]. A further option was the engineered *m*-DAPDH from *B. sphaericus*; co-expression in *E. coli* cells of this DAADH with glucose dehydrogenase from *Gluconobacter oxidans* allowed the production of D-5,5,5-trifluoronorvaline, an intermediate in the synthesis of the γ-secretase inhibitor BMS-708163 [29] from the corresponding α-keto acid: the preparation reached the 50 kg scale, with >98.6% ee and 89% yield.

### 3.2. Stereoinversion of L-Amino Acids

LAAD and DAADH were used for the stereoinversions of L-AAs into D-AAs, Figure 4. L-Phenylalanine was deaminated to phenylpyruvic acid using *E. coli* cells expressing PmLAAD, followed by stereoselective reductive amination with recombinant *m*-DAPDH from *S. thermophilum* (both the wild-type and the H227V variants) to produce D-phenylalanine; formate dehydrogenase was used to regenerate NADPH. D-phenylalanine, as well as the D-enantiomer of glutamate, homophenylalanine and 2-, 3-, or 4-chlorophenylalanine were produced in quantitative yield and with >99% ee (using a 10 mM solution of the L-enantiomer, 20 mM sodium formate and 1 mM NADP^+^) [30]. D-Phenylalanine and D-arylalanines are components of natural products such as antibacterial peptides (i.e., fungisporin, polymyxin, gramicidin) and alkaloids, and ingredients of antidiabetics (i.e., nateglinide), anti-inflammatory formyl peptide receptor 1 antagonists, anticoagulants, cyclic plasmin inhibitors and melanocortin 4 receptor agonists for the treatment of obesity. On this side, an engineered DAADH obtained by mutagenesis of *m*-DAPDH from *Corynebacterium glutamicum* (harboring the N150L/D154G/T169I/R195M/H244E substitutions) was coupled with PmLAAD and the NADPH recycling system made of glucose oxidase/glucose [31]. The reaction resulted in excellent enantioselectivity (>98%) and good yields (69–93%) at the preparative scale (100 mg) across a broad range of D-arylalanines/D-phenylalanine derivatives and using cheaper L- or D,L-amino acid substrate mixtures.

## 4. Hydantoinase Process

The “hydantoinase process” is an industrial multi-enzymatic system in which hydantoin racemase catalyzes the racemization of L- or D-hydantoin, D-hydantoinase hydrolyzes D-hydantoin to the corresponding d-*N*-carbamoyl-amino acid, then converted into D-AAs by D-carbamoylase (Figure 5). Various D-AAs have been successfully synthesized at industrial scale by this dynamic kinetic resolution cascade process, with high yield and enantioselectivity; for a review, see [32]. D-Valine, an important organic chiral precursor used for the synthesis of agricultural pesticides, semisynthetic veterinary antibiotics and pharmaceutical drugs, has been successfully produced by the hydantoinase process, reaching 88% of conversion in 48 h starting from 50 g/L of D,L-5-isopropylhydantoin, see [33]. In order to bypass some limitations in this promising strategy, the D-carbamoylase from *Arthrobacter crystallopoietes* was recently identified: it is more compatible with the hydantoinase process conditions, is produced in a soluble form and shows a substrate preference for aromatic carbamoyl compounds [34]. By combining this D-carbamoylase with *Arthrobacter aurescens* hydantoin racemase and *Agrobacterium tumefaciens* D-hydantoinase, 80 mM L-indolylmethylhydantoin was converted into D-tryptophan (99.4% yield and >99.9% ee) within 12 h and with a productivity of 36.6 g/L/day (at 0.5 L scale). D-Tryptophan is used to produce important drugs in the treatment of erectile dysfunction or pulmonary arterial hypertension (i.e., contryphans and tadalafil), peptides for the treatment of dermatitis (i.e., tyrocidines C and D, or Thymodepressin) and as sweetener for food industry.

## 5. Ammonia Lyases

Ammonia lyases (E.C. 4.3.1.X) are enzymes that catalyze the reversible deamination of unsaturated or cyclic amino acids. Phenylalanine ammonia lyase (PAL, EC 4.3.1.24, belonging to class-I lyase family), that catalyzes the deamination of L-phenylalanine to produce cinnamic acid, a precursor of several secondary metabolites in plants, bacteria and fungi, represents the most biotechnological relevant member [35]. Interestingly, PAL’s activity relies on a peculiar electrophilic cofactor (4-methylideneimidazol-5-one, MIO) derived from the condensation of three residues at the active site (Ala167, Ser168, and Gly169 in *Anabena variabilis* PAL, AvPAL). The advantages of PAL’s exploitation in biocatalysis are: no requirement for cofactor supplementation, no recycling systems and, in most cases, an absolute regioselectivity, see [36].

### 5.1. Production of D-Amino Acids by Enantioselective Deamination

The exploitation of PAL to produce optically pure D-AAs follows three major biocatalytic approaches. In the enantioselective deamination approach, the enzyme has been used (mainly as a whole cell) to remove the L-enantiomer from a racemic mixture to leave the unreacted D-AA. The generated α-keto acid can be easily separated from the D-AA by precipitation or column separation [37]. On this side, the I460V variant of *Petroselinum crispum* PAL (PcPAL) was recently used in preparative scale biotransformation to efficiently produce D-*m*-(trifluoromethyl)phenylalanine (39% yield, 93% ee) and D-*p*-methylphenylalanine (49% yield, 95% ee), intermediates for the synthesis of inhibitors of lysine methyltransferase 7 and prolyl isomerase (anti-cancer drugs) [38,39] (Figure 6A). A recent evolution of this approach is represented by the immobilization of the biocatalyst with the aim to improve stability and facilitate recovery. PAL from *Rhodotorula glutinis* immobilized on a silica support has been used in a recirculating packed-bed reactor to produce optically pure D-phenylalanine from the racemic mixture [40]. The resolution process was scaled up to 25 L of 100 mM racemic amino acid obtaining a maximal productivity of 7.2 g/L/h D-phenylalanine (≈95% yield and 99% ee). The reactor could be operated 16 consecutive times without significant decrease in performances. PALs have recently been immobilized also on advanced supports such as nanomaterials; PAL from parsley has been immobilized on single-walled carbon nanotubes and used in a packed-bed continuous-flow microreactor for the kinetic resolution of racemic 2-amino-3-(thiophen-2-yl) propanoic acid. The enzyme could be used for seven consecutive cycles maintaining a conversion yield >42% [41]. PcPAL was immobilized on magnetic nanoparticles allowing its use in a Lab on a Chip magnetic system, a reliable tool for microscale biotransformation and automated screenings of substrates [42]. This system demonstrated the ability of PcPAL (subsequently confirmed by a preparative biotransformation) to catalyze the enantioselective deamination of D,L-propargylglycine, which represents the sole non-cyclic relevant substrate of PALs (Figure 6B) [43].

### 5.2. Production of D-Amino Acids by Enantioselective Hydroamination

Under high ammonia concentration, the deamination reaction catalyzed by PALs is reversible, leading to the non-reductive hydroamination of substituted cinnamic acids (aryl α,β unsaturated carboxylic acids) by an anti-Michael addition of ammonia. The stereochemistry of this latter reaction is very high for the production of the L-AA (ee for the L-enantiomers is usually >99%), but when the aromatic ring of cinnamic acid is substituted with an electron abstracting group (e.g., *p*/*o*-NO_2_ or pyridyl-alanines), a significant fraction of the product is represented by the D-enantiomer. This is due to ability of electron-deficient structures to stabilize a negative charge at the benzylic position. Investigation of the addition reaction demonstrated that the formation of D-AAs by PAL-catalyzed hydroamination occurs via a MIO-independent pathway which is not enantioselective [44]. The best ee toward the D-AA was obtained using *R. glutinis* PAL on *p*NO_2_-phenylalanine (corresponding to 55% of the D-AA). Similar results have been obtained also using PALs from different sources, i.e., AvPAL or PcPAL (Figure 7A) [44].

### 5.3. Production of D-Amino Acids by One-Pot Deracemization Reaction

With the aim to achieve a full production of D-AAs, the (almost) asymmetric hydroamination of substituted cinnamic acids to produce non-natural D-phenylalanines catalyzed by AvPAL was coupled to the enantioselective deamination reaction catalyzed by LAAD. PmLAAD, expressed in *E. coli* and used as a whole-cell biocatalyst, stereospecifically deaminated the L-enantiomer of the substituted phenylalanine (produced by PAL) yielding the corresponding imino acid that, in turn, was asymmetrically reduced back to the racemic amino acid by ammonia-borane (NH_3_BH_3_). The iteration of these two reactions lead to the accumulation of the D-AA. Using the wild-type AvPAL (as a whole-cell biocatalyst), *p*NO_2_-cinnamic acid was converted into *p*NO_2_-D-phenylalanine (71% yield and 96% ee) (Figure 7B) [45]. The enzymatic activity of AvPAL has been enhanced through a semi-rational (site-saturation mutagenesis) approach; the best variant (H359Y) showed a ≈3.5-fold higher activity with no improvement in stereospecificity toward the D-product. Using this variant, an up to 80% yield with an ee between >98% was reached using different substituted cinnamic acids as substrate. Since the deracemization step involving PmLAAD exhibits a very high efficiency, the cascade bioconversion could also be used for the full stereoinversion of substituted L-phenylalanines [45].

A different variant of AvPAL (N347A), possessing a 2.3-fold higher D-enantioselectivity in comparison with the wild-type enzyme, was active toward aromatic amino acids possessing strong electron-withdrawing substituents (e.g., *p*NO_2_-phenylalanine) [45]. The N347A AvPAL variant was used with PmLAAD in a one-pot deracemization bioprocess; the best results have been reached with *m*F- and *p*NO_2_-phenylalanines (91% and 96% yield, respectively, and >99% ee). Interestingly, the N347A AvPAL variant was also active on phenylalanines possessing strongly electron-donating groups (e.g., *m*CH_3_- or *p*OH-phenylalanines), although the conversion rates were much lower (23% and 26% yield, respectively) [46].

The main drawback of this approach is the cost associated to the large excess (40 equivalents) of ammonia-borane as the chemical reducing agent requested to avoid the production the α-keto acid side product (Figure 7B). In order to overcome this limitation, the asymmetric chemical reduction of the imino acid intermediate was replaced by the enantioselective amination of the α-keto acid to the D-AA catalyzed by DAAT [47]. This approach was used to produce D-(2,4,5-trifluoro-phenyl)alanine, the precursor of the antihyperglycemic drug sitagliptin (approved by FDA for the treatment of type 2 diabetes) [47]. The whole biocatalytic process started from trifluoro-cinnamic acid, prepared by one-step condensation between trifluorobenzaldehyde and malonic acid, which was converted to L-trifluoro-phenylalanine by AvPAL (in 2.5 M NH_3_/CO_2_ buffer) and, in turn, fully stereoinverted to the D-AA by the coupled oxidative deamination of the L-AA to its corresponding α-keto acid and the subsequent enantioselective amination to the D-enantiomer catalyzed by DAAT (using D-aspartate as the amino donor). Both LAAD and DAAT were added as whole-cell biocatalysts. Since high ammonia concentrations are detrimental to the enzymes involved in the deracemization, the reaction mixture has been diluted with an equal volume of water after the hydroamination catalyzed by PAL before adding LAAD and DAAT [47] (Figure 7C). As a proof of concept, a complete biotransformation of 50 mM trifluoro-cinnamic acid to D-trifluoro-phenylalanine was reached in a few hours. The product was isolated incubating the reaction with recombinant bovine D-aspartate oxidase (whole cells) to eliminate D-aspartate, followed by ion exchange purification affording a 67% yield. In addition, inexpensive ammonium carbamate (that can be easily removed by sublimation) [37] can be used as ammonium source, rendering the whole process more economically and virtually feasible within an industrial setting [47].

## 6. L-Amino Acid Oxidases and L-Amino Acid Deaminases

L-amino acid oxidase (LAAO, EC 1,4.99.B3) and L-amino acid deaminase (LAAD, EC 1.4.99.B3) catalyze the enantioselective oxidation of L-amino acids to their imino form that spontaneously deaminate forming the corresponding α-keto acid. The main difference between the two enzymes is the electron acceptor, that is molecular oxygen (with production of hydrogen peroxide) in the case of LAAO while, in the case of LAAD, it is a component of the bacterial membrane electron-transport chain (with the final production of water) [48,49,50].

### 6.1. Production of D-Amino Acids by Enantioselective Deamination/Deracemization Using L-Amino Acid Oxidase

The availability of recombinant microbial D-amino acid oxidase variants with improved properties allowed the setup of several protocols to produce optically pure natural or unnatural L-AAs [51,52]. Unfortunately, the unavailability of suitable recombinant LAAOs, prevented the effective application of similar approaches to produce optically pure D-AAs [52]. Recently, LAAO from the actinomycete *Lechevalieria aerocolonigenes* (RebO) was used in the synthesis of halogenated derivatives of D-tryptophan using a bienzymatic cascade system (Figure 8A). Halogenated L-tryptophan derivatives were prepared from L-tryptophan by tryptophan halogenases specific for position 5, 6 or 7 of the indole side chain. The L-enantiomer was then stereoinverted by the combined use of L-Enantioselective RebO and the non-enantioselective reducing agent ammonia-borane. This system allowed reaching ee ≈ 100% for six out of eight tested substrates. In a quantitative conversion (1 mM substrate), halogenation of tryptophan to L-5-Br-tryptophan was achieved in 3−5 h by halogenase in the form of cross-linked enzyme aggregate CLEA (comprising also auxiliary enzymes for cofactor regeneration). L-5-Br-tryptophan was then fully stereoinverted to the D-AA (49% yield, 92% ee) after 24 h of incubation with RebO (as a crude lysate) and ammonia-borane. Similar results were obtained for D-7-Br-tryptophan (44% yield, >98% ee). Drawbacks of this system are the need for a very high stoichiometric excess of the reducing agent (100 equivalents), the need of catalase to remove H_2_O_2_ produced by RebO, and the generation of the side product α-Hydroxy acid derivative (≈10% of the total product) due to chemical reduction of the α-keto acid [53]. It is generally accepted that during evolution, generalist enzymes acquired a narrower enzyme specificity, thus, ancestral variants of native enzymes could possess a broader substrate specificity [54]. According to this concept, Nakano and colleagues produced a potential ancestral variant (AncLAAO) of LAAO from *Pseudoalteromonas piscicida.* AncLAAO showed significant activity towards seven proteinogenic L-AAs (relative activity >20%, the best being L-glutamine and L-methionine) and on several substituted tryptophans and phenylalanines. AncLAAO was employed to deracemize four substituted phenylalanines (2-fluoro-, 3-fluoro-, 4-nitro-, and 3,4,5-trifluoro-phenylalanine) and phenylglycine at a preparative scale (100 mL of 10 mM racemate corresponding to 151–210 mg of substrate, in a reaction mixture containing 150 mM NH_3_BH_3_, 200 U catalase, at 30 °C, Figure 8B). The reaction required a 10-fold higher amount of the biocatalyst in the case of phenylglycine (70 mg vs. 7 mg required for the deracemization of phenylalanines). A >99% ee was obtained for the phenylalanine derivatives (after a 24 h reaction time) whereas 84% ee was for phenylglycine. One main advantage in using AncLAAO is represented by the very high recombinant expression level in *E. coli*: 50 mg/L fermentation broth [55].

### 6.2. Production of D-Amino Acids by Enantioselective Deamination/Deracemization Using L-Amino Acid Deaminase

A promising alternative to LAAOs as biocatalyst for the enantioselective oxidative deamination of L-AAs is represented by LAADs [48]. The main advantages are the broad substrate specificity and the absence of H_2_O_2_ production, this resulting in a lower toxicity during its recombinant expression in *E coli*, an increased stability of the enzyme and reaction products during the reaction, and the elimination of catalase from the reaction mixture. LAADs can be used as whole-cell biocatalysts or in the pure form, the latter requiring the addition of purified bacterial membranes containing a suitable electron acceptor. On this side, it has been recently demonstrated that the bacterial membranes can be replaced by the artificial electron acceptor represented by 0.8 mM phenazine methosulfate and the anionic detergents SDS (0.1 mM) [50].

One of the first examples reported the application of LAAD (as whole *E. coli* recombinant cells) to produce pure D-AAs from the racemic mixture; after the oxidation of the L-AA to the imino acid intermediate, the latter compound was non-enantiomerically reduced by 40 equivalents of ammonia-borane. This approach has been used to produce several proteinogenic (e.g., D-leucine or D-methionine) or non-proteinogenic (e.g., *o*-benzyl-D-serine or D-norleucine,) optically pure apolar D-AAs (the sole exception being histidine) obtaining reaction yields ranging from 64% to 99% and ee up to 99% in most cases [56]. D-2-amino-3-(7-methyl-1 *H*-indazol-5-yl)propanoic acid (a drug building block) has been produced coupling the reaction of PmLAAD in combination with a DAAT from *Bacillus thuringiensis*: a 92% yield (79% after product purification) with a >99% ee was achieved. Notably, 1.79 kg of product were isolated (62% isolated yield and 98.6% ee) on a pilot scale using a commercial DAAT. The main drawback of this approach is the need of D-alanine as the DAAT amino donor and of an additional pyruvate removal reaction requiring NADH recycling [57]. Recently, PmLAAD has been expressed together with two additional enzymes in an artificial enzyme cascade in *E. coli* with the aim to produce benzyl alcohol analogs. This approach demonstrated the ability of PmLAAD to produce optically pure D-*m*-fluoro-phenylalanine from the corresponding racemic mixture [58]. In addition, a 4-point variant of this enzyme has been exploited to produce D-phenylalanine, exhibiting a conversion of 49.5% after 7 h [59]. For recent applications of LAAD from *Proteus* to produce D-AA by enzymatic cascades, see also paragraph 2 [45,46]. LAAD from *P. mirabilis* (originally termed L-amino acid oxidase) has been recently immobilized on Ca-alginate beads and employed for producing D-phenylalanine following the deamination of the L-AA [60].

The recent setup of a simple and efficient one-step purification protocol of a deletion variant of PmaLAAD (lacking the native *N*-terminal signal peptide) and the solution of its 3D structure boosted its exploitation as a purified biocatalyst [49]. PmaLAAD was used in deracemization of proteinogenic hydrophobic amino acids (e.g., L-phenylalanine, L-leucine, L-methionine, and L-tryptophan), non-proteinogenic amino acids (L-3,4-dihydroxyphenylalanine) and even unnatural amino acids such as substituted alanines (e.g., 3-pyridyl- or naphthyl-alanines). Starting from the racemic mixture, almost 100% of conversion of the L-AA into the corresponding α-keto acid was obtained, leaving only the D-form in solution. The coupling of the reaction with a non-enantiomeric in situ reduction step (by 5 equivalents *tert*-butylamine) allowed the full stereoinversion of 12.5 mM L-4-NO_2_-phenylalanine into the corresponding D-enantiomer in 2 h (99% ee) (Figure 9A) [61]. Based on the available 3D structure of PmaLAAD, variants showing improved activity towards substituted naphthylalanines were generated by site-saturation mutagenesis (Figure 9B). The triple variant F318A/V412A/V438P PmaLAAD was employed for the full deracemization of a solution of 1.2 mM D,L-1-naphthylalanine, achieving >90% conversion of the L-enantiomer in a reaction time ∼7.5-fold lower than the wild-type enzyme [62]. For additional examples of the exploitation of LAAOs and LAAD in multi-enzymatic cascades, see above.

## 7. Conclusions

The advances in the past five years, as compared to the previously reported methods for D-AAs production, focused on the intrinsic atomic economy and the low environmental impact of the process. Some of the most relevant successes were:-the full conversion of enantiomeric solutions of amino acids into the D-enantiomer;-the production of a variety of unnatural D-AA derivatives by means of novel enzymes, both isolated from natural sources (e.g., LAADs) and generated by protein engineering (DAADHs and ARTA);-the optimization of the hydantoinase process by the identification of well-suited enzymes that rendered it one of the most promising strategies to achieve economic and enantioselective synthesis of D-AAs;-the synthesis by multi-enzymatic processes of phenylalanine and tryptophan, and their derivatives, is now a reality [12,34,53];-the establishment of an attractive synthetic method for D-2-aminobutyric acid by the reductive amination of 2-oxobutyric acid by DAADH [25];-the enantioselective deamination of L-AAs by enzymes immobilized on magnetic nanoparticles in a microfluidic reactor [43].

As stated by Stefano Servi, “the interconversion of functional groups is the main objective of biocatalysis, and systems organizing a series of enzymes to achieve multistep reactions represent the advanced target of enzymatic catalysis for organic synthesis” [63]. The plethora of enzymes related to D-AAs allow now to consider them as an appropriate example of “systems biocatalysis”, i.e., in vitro pathways of alternative/exchangeable enzymes that allow the generation of an artificial metabolism for synthetic purposes. 

## Figures and Tables

**Figure 1 ijms-21-03206-f001:**
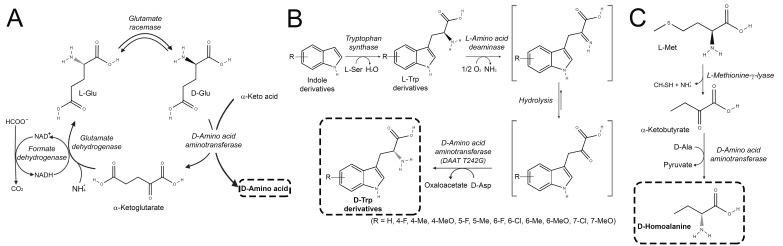
Use of aminotransferases in production of D-AAs. Synthesis of D-AAs from the corresponding α-keto acids and ammonia by coupling: (**A**) four enzymes, namely D-amino acid aminotransferase, glutamate racemase, glutamate dehydrogenase and formate dehydrogenase [11]; (**B**) tryptophan synthase from *S. enterica,* L-amino acid deaminase from *P. myxofaciens* and T242G variant of D-aminotransferase variant from *Bacillus* sp. YM-1 for the synthesis of D-tryptophan derivatives [12]; (**C**) L-methionine γ-lyase from *F. nucleatum* and D-amino acid aminotransferase from *Bacillus* sp. to convert L-methionine into D-homoalanine [13].

**Figure 2 ijms-21-03206-f002:**
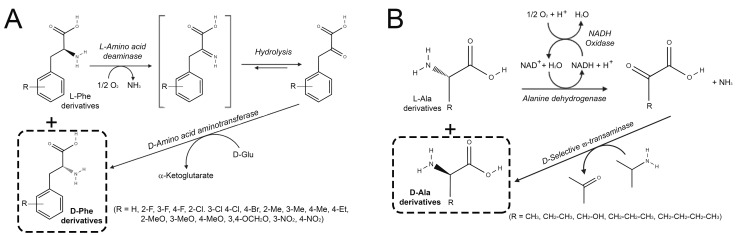
Stereoinversion and deracemization of: (**A**) phenylalanine derivatives by means of L-amino acid deaminase and a D-selective transaminase [18]; (**B**) alanine derivatives by alanine dehydrogenase from *B. subtilis,* NADH oxidase and the engineered D-selective ω-transaminase from *Arthrobacter* sp. [19].

**Figure 3 ijms-21-03206-f003:**
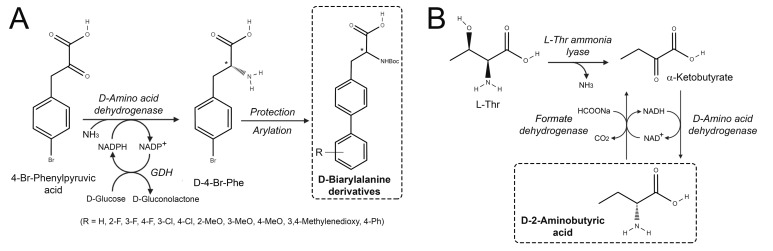
Use of engineered D-amino acid dehydrogenases in production of D-AAs. Synthesis of D-AAs by: (**A**) direct conversion by engineered D-amino acid dehydrogenase of 4-Br-phenylpyruvic acid into D-4-Br-phenylalanine, subsequently coupled with a panel of arylboronic acids to give D-biarylalanine derivates [21]; (**B**) the multi-enzymatic system made of *E. coli* L-threonine ammonia lyase, D-amino acid dehydrogenase and formate dehydrogenase converted L-threonine into D-2-aminobutyric acid [25].

**Figure 4 ijms-21-03206-f004:**
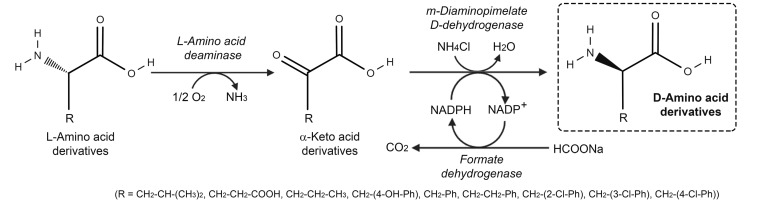
One-pot stereoinversion of L-AAs by *P. mirabilis* L-amino acid deaminase and recombinant *meso*-diaminopimelate D-dehydrogenase from *S. thermophilum*; formate dehydrogenase was used to regenerate NADPH.

**Figure 5 ijms-21-03206-f005:**
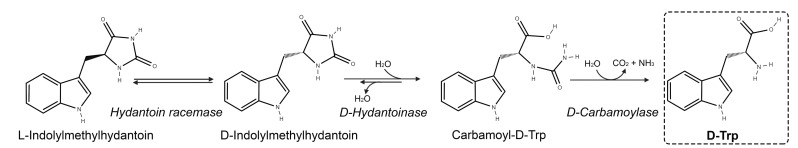
The application of the hydantoinase process to the synthesis of D-tryptophan. In this case, hydantoin racemase from *A. aurescens* was coupled to D-hydantoinase from *A. tumefaciens* and D-carbamoylase from *A. crystallopoietes* [34].

**Figure 6 ijms-21-03206-f006:**
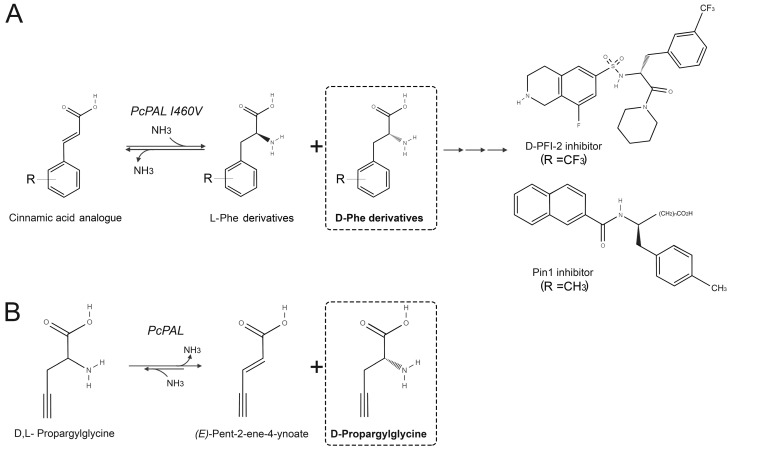
The application of PcPAL in D-AAs synthesis. (**A**) Resolution of substituted D-phenylglycines by the I460V PcPAL variant [39]. (**B**) Preparation of enantiopure D-propargylglycine by enantioselective deamination of L-propargylglycine catalyzed by PcPAL immobilized on magnetic nanoparticles [43].

**Figure 7 ijms-21-03206-f007:**
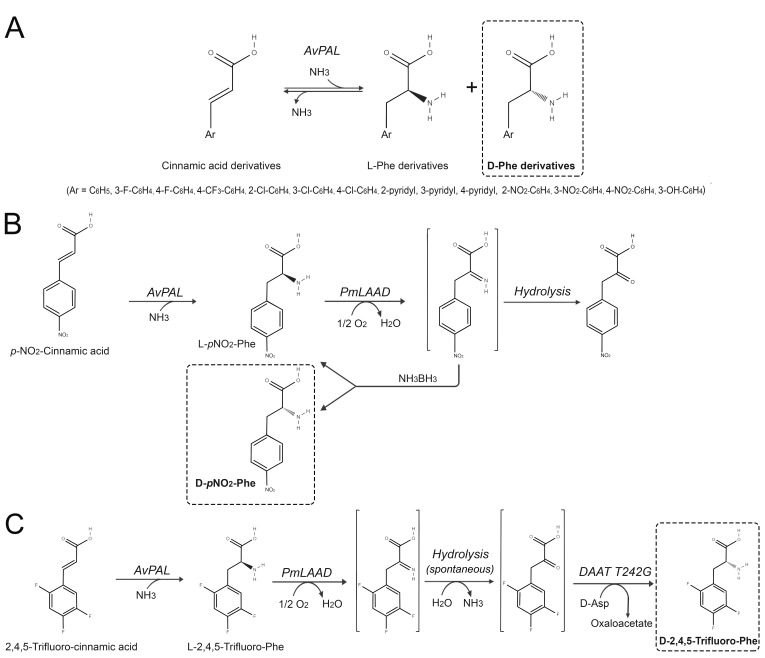
The application of AvPAL in D-AAs synthesis. (**A**) Production of substituted D-phenylalanines by enantioselective hydroamination of cinnamic acid derivatives using AvPAL [44]. (**B**) Production of *p*NO2-phenylalanine from the corresponding racemate employing AvPAL, *P. mirabilis* D-amino acid deaminase, and amine borane [45]. (**C**) Chemo-enzymatic pathway for the synthesis of optically pure D-(2,4,5-trifluorophenyl)alanine from 2,4,5-trifluoro-cinnamic acid generated from trifluorobenzaldehyde using the DAAT T242G variant from *Bacillus* sp. YM-1, PmLAAD and AvPAL [47].

**Figure 8 ijms-21-03206-f008:**
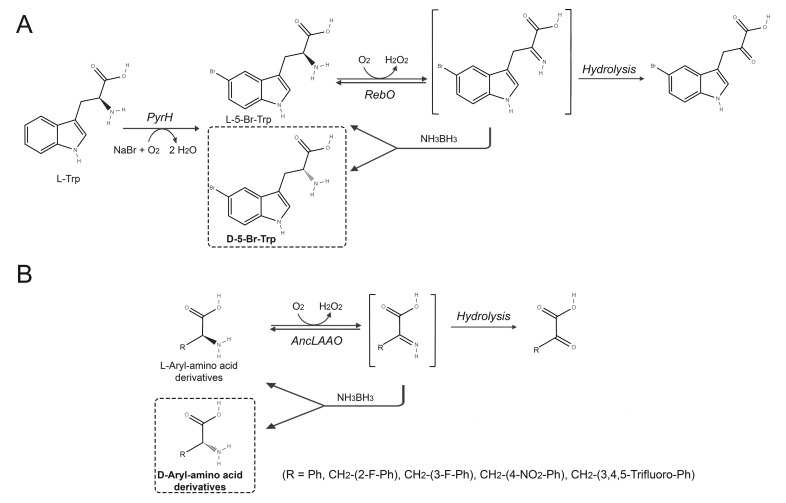
Production of D-AAs by L-amino acid oxidase. (**A**) Two-step biocatalytic synthesis of D-5-Br-tryptophan using a one-pot approach by combining tryptophan 5-halogenase from *Streptomyces rugosporus* with RebO, without intermediary purification. Iteration of enantioselective oxidation of the L-enantiomer by RebO and subsequent chemical reduction into the racemate induced accumulation of the D-AA [53]. (**B**) Application of ancestral variant of LAAO (AncLAAO) in the deracemization of substituted phenylalanines and phenylglycine [55].

**Figure 9 ijms-21-03206-f009:**
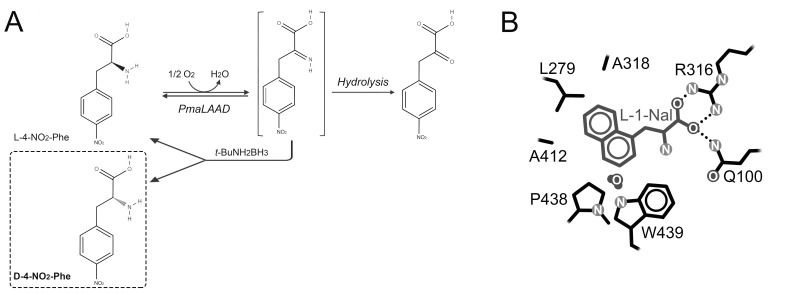
Biocatalytic application of PmaLAAD. (**A**) Stereoinversion of 12.5 mM L-4-NO_2_-phenylalanine, using 62.5 mM borane *tert*-butylamine, 0.1 mg PmaLAAD/mL, at 25 °C and pH 7.5 [61]; (**B**) schematic structure of the active site of the triple variant F318A/V412A/V438P PmaLAAD with L-1-naphthylalanine modelled as substrate [62].

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
