# Peer review of "Advances in Enzymatic Synthesis of D-Amino Acids"

_ijms, 2020, doi:10.3390/ijms21093206_

Round 1
Reviewer 1 Report
Please find my comments about the results and the presentation below:
- Line 62-63: Please add first author’s name before the publication reference “[12]”.
- Figures 1, 4, 5: This figures are unreadable. Authors must divide this pictures into several pieces.
- Lines 160-162:” At 50 mL reaction scale, 200 mL L- threonine and 300 mM sodium formate generated D-2-aminobutyric acid in 20 h, with > 90% yield and > 99% ee, with no need for external ammonia supplement [25].” This sentence is not clear. Could authors corrected it?
- Line 171: Please add the full name of this microorganism.
- Lines 381:” A > 99% ee was obtained for the phenylalanine derivatives (after a 24 h reaction time) vs. 84% ee for phenylglycine” This sentence is not clear. Could authors corrected it?
Author Response
- Line 62-63: Please add first author’s name before the publication reference “[12]”.
First author’s name added
- Figures 1, 4, 5: This figures are unreadable. Authors must divide this pictures into several pieces.
We agree with the referee’s observation. The original organization of figures was meant to keep the number of individual figures as low as possible. Since this issue has been raised also by the other reviewers, in the present version of the manuscript some figures that contained multiple panels have been divided.
- Lines 160-162:” At 50 mL reaction scale, 200 mL L- threonine and 300 mM sodium formate generated D-2-aminobutyric acid in 20 h, with > 90% yield and > 99% ee, with no need for external ammonia supplement [25].” This sentence is not clear. Could authors corrected it?
We apologize for the mistake: “200 mL” was corrected to “200 mM”.
- Line 171: Please add the full name of this microorganism.
Full name of microorganism has been added.
- Lines 381:” A > 99% ee was obtained for the phenylalanine derivatives (after a 24 h reaction time) vs. 84% ee for phenylglycine” This sentence is not clear. Could authors corrected it?
“vs.” has been replaced with “whereas” as in the cited paper.
Reviewer 2 Report
Comments for the Authors
In this review, the authors reported “Advances in enzymatic synthesis of D-amino acids”. D-amino acids are key intermediates in the synthesis of several important pharmaceuticals, food and cosmetic industries. Also, enzymatic synthesis has several advantages over conventional chemical synthesis.
- The review is well written and structured
- Instead of mentioning Figure 1A, 2 A right and left. The authors should segregate the figures clearly into 1A, 1B, 1C and so on, to have a clear understanding
- It would be nice if the authors include some more examples in the review, particularly in the section D-amino acid dehydrogenases
I believe this review is suitable for publication in International Journal of molecular sciences
Author Response
- Instead of mentioning Figure 1A, 2 A right and left. The authors should segregate the figures clearly into 1A, 1B, 1C and so on, to have a clear understanding
We agree with the referee’s observation. The original organization of figures was meant to keep the number of individual figures as low as possible. Since this issue has been raised also by the other reviewers, in the present version of the manuscript some figures that contained multiple panels have been divided.
- It would be nice if the authors include some more examples in the review, particularly in the section D-amino acid dehydrogenases
Accordingly to the referee’s suggestion, we performed a new check of literature concerning “D-amino acid dehydrogenases” and “D-amino acids synthesis”. We introduced one recent publication that was not originally included into the manuscript (Akita, H.; Hayashi, J.; Sakuraba, H.; Ohshima, T. Artificial thermostable D-amino acid dehydrogenase: creation and application. Front. Microbiol. 2018, 9, 1760), and a related sentence. No additional papers have been identified related to this topic.
Reviewer 3 Report
Dear Authors
The review is a well prepared survey on recent findings in the field of D-AA production.
minor comments only:
L78, L132, L157 ... "sp." change style to non-italic
L92 "R" change style to italics ... check for all R/S throughout manuscript.
Fig1 is pretty packed and low in resolutions; it might be worth to separate it into more than 1 figure.
L192 ? what is "D.L" ?
L238 "mainly"
throughout the manuscript streamline para/ortho or p/o ... any way it should be italics
L370 ... enter line break after figure legend
Author Response
- L78, L132, L157 ... "sp." change style to non-italic
Style was changes accordingly.
- L92 "R" change style to italics ... check for all R/S throughout manuscript.
Style was changes accordingly.
- Fig1 is pretty packed and low in resolutions; it might be worth to separate it into more than 1 figure.
We agree with the referee’s observation. The original organization of figures was meant to keep the number of individual figures as low as possible. Since this issue has been raised also by the other reviewers, in the present version of the manuscript some figures that contained multiple panels have been divided.
- L192 ? what is "D.L" ?
Corrected to “D,L”.
- L238 "mainly"
Corrected.
- throughout the manuscript streamline para/ortho or p/o ... any way it should be italics
“p/o” were changed to italics
- L370 ... enter line break after figure legend
Line break added.
Round 2
Reviewer 1 Report
I don't have more questions to authors